# Participation in microfinance based Self Help Groups in India: Who becomes a member and for how long?

Danish Ahmad[1,2]*, Itismita Mohanty[1], Laili Irani[3], Dileep Mavalankar[2], Theo Niyonsenga[1]

1 Health Research Institute, Faculty of Health, University of Canberra, Canberra, Australia, 2 Public Health Foundation of India, and Indian Institute of Public Health Gandhinagar (IIPHG), Gandhinagar, Gujarat, India, 3 Population Council, New Delhi, India

* drdanish.research@gmail.com

## Abstract

### Introduction

Microfinance is a widely promoted developmental initiative to provide poor women with affordable financial services for poverty alleviation. One popular adaption in South Asia is the Self-Help Group (SHG) model that India adopted in 2011 as part of a federal poverty alleviation program and as a secondary approach of integrating health literacy services for rural women. However, the evidence is limited on who joins and continues in SHG programs. This paper examines the determinants of membership and staying members (outcomes) in an integrated microfinance and health literacy program from one of India's poorest and most populated states, Uttar Pradesh across a range of explanatory variables related to economic, socio-demographic and area-level characteristics.

### Method

Using secondary survey data from the Uttar Pradesh Community Mobilization project comprising of 15,300 women from SHGs and Non-SHG households in rural India, we performed multivariate logistic and hurdle negative binomial regression analyses to model SHG membership and duration.

### Results

While in general poor women are more likely to be SHG members based on an income threshold limit (government-sponsored BPL cards), women from poorest households are more likely to become members, but less likely to stay members, when further classified using asset-based wealth quintiles. Additionally, poorer households compared to the marginally poor are less likely to become SHG members when borrowing for any reason, including health reasons. Only women from moderately poor households are more likely to continue as members if borrowing for health and non-income-generating reasons. The study found that an increasing number of previous pregnancies is associated with a higher

**Data Availability Statement:** All study files are available from the Population Council Dataverse database (https://doi.org/10.7910/DVN/KSRVL).

**Funding:** The paper is part of the PhD research of the first author(Danish Ahmad) who is supported by the University of Canberra's Higher Degree by Research scholarship for his PhD. Open access funding for the publication is provided by the Bill and Melinda Gates Foundation who also funded the program evaluated in the paper. The funders had no role in study design, data collection and analysis, decision to publish, or preparation of the manuscript.

**Competing interests:** The authors have declared that no competing interests exist.

membership likelihood in contrast to another study from India reporting a negative association.

## Conclusion

The study supports the view that microfinance programs need to examine their inclusion and retention strategies in favour of poorest household using multidimensional indicators that can capture poverty in its myriad forms.

## Introduction

Microfinance (MF) is widely promoted as a developmental initiative to provide low-income populations with affordable financial services. Reflecting a gradual shift in perception that considered the poor to be previously 'un-bankable', microfinance is now an established sector with an investment portfolio over USD 114 billion and a reach of more than 139 million clients across 60 countries [1, 2]. MF clients are likely to be marginalised rural poor populations, with minimal or no landholding, low levels of literacy, and other groups without a bank account; which in 2017, still comprised 1.67 billion people or 22 per cent of the global population [2, 3]. This is relevant to populations in South Asia, where the highest number of the world's severely poor living and where more than 60 per cent of global microfinance borrowers are also found, the majority being women [2, 3].

While still widely promoted by global institutions as a poverty alleviation *approach*, unregulated microfinance practice in the past has harmed clients by promoting 'over-indebtedness' as a result of repeated loan taking with negative implications for livelihoods[4, 5]. Therefore, there needs to be a renewed focus on what microfinance intends to achieve in terms of social development and identify barriers in its progress. A 2012 systematic review of microfinance's impact in Sub Saharan Africa, for example, reviewed 15 studies and found that 'in some cases, microfinance increased poverty, reduced levels of children's education and disempowered women' [4]. Some criticism of microfinance found that 'microfinance is largely *antagonistic* to sustainable economic and social development, and consequently to sustainable poverty reduction' [5, 6].

These views, however, need to be placed in the broader debate and assumptions of what qualifies as a successful MF program. Another aspect that intensifies the debate is that specific MF experiences from different regions, like different community programs, are limited in their scope of generalisability, as program successes and failures from one region to another may not be comparable given regional and cultural dissimilarities [7, 8]. The layout of the paper is as follows; the first section provides context on current microfinance evidence with reference to a specialised microfinance model involving women only Self-Help Groups (SHG) popular in South Asia especially in India. The next section describes the methodology, project context and survey sampling followed by details of the statistical analysis. Subsequently, the results are separately presented for the determinants of SHG membership, and duration of the membership. Lastly, the paper discusses the key determinants for both membership and duration and provides policy and program implication of this research. This study addressed a significant research gap on who becomes a member of the microfinance based SHGs in rural India and more importantly, what determines the duration of their SHG membership. The study analysed membership profiles, especially among gradients of the poor and the poorest, using available microfinance data from rural India. Additionally, the study analysed who

benefits from SHGs and what drives women in rural India to join and continue as members in women-only groups.

## Microfinance: Evidence so far

This study focuses on a specialised microfinance model involving women only Self-Help Groups in rural India with an additional layer of health intervention. The SHG model is popular in South Asia [9]. India adopted the SHG program as part of its national poverty alleviation program and is home to a substantial, large poor population.

## Self-help Group (SHG): Basic context

The SHG model in India evolved from early microfinance experiences of the Grameen Bank in Bangladesh, and developmental program(s) in South India in the early 1970-80s [3, 10]. These formative schemes showed the benefits of providing small loans to disadvantaged women, who then reported higher loan repayment rates but also higher utilisation of loans towards household wellbeing [3, 4, 10]. At its core, the SHG model comprises 10–15 poor adult women (clients) organised into groups with members mostly from a similar background or socio-economic status [2, 10]. Targeting clients for inclusion into SHGs is mostly done by governments and Non-Governmental Organisations (NGO), often through a local community facilitator who seeks to identify the poorest households with community assistance and use of locally adapted but objective criteria defining poverty [10]. SHGs specifically seek to target income poor women, especially the poorest amongst them, who sit at the bottom of the poverty pyramid, where higher socio-economic inequalities and deprivations cluster [2, 3, 10]. Member selection targeting the poor, hence, is a crucial consideration for the success of such programs.

For SHG to function at a minimum, adult women representing households, are required to broadly follow certain principles(regular meetings -up to four times /month, regular savings, internal lending, regular repayment and record-keeping) [10]. In India, rural public banks mainly use these indicators to grade SHG functioning after a new group is formed over three to six months to assess their suitability to receive loans, and the amount of bank loan or credit that can be dispensed [10]. Rural banks typically offer SHGs incrementally higher loan amounts or credit limits depending on how well the group continues to perform and return previously borrowed loans which are a collective responsibility of group members [10]. SHG membership is thought to foster a sense of group solidarity and cohesion in supporting each other to repay loans and also function as a broader 'peer network' for women [11, 12]. Therefore, member characteristics are vital in determining not only how well SHGs function but also for how long they continue to function. The available evidence on membership determinants comes from a few studies from rural India that mainly find that while members are poor, they tend to be at or just above poverty thresholds; the poorest are often excluded from membership [13–15].

However, limited studies have reported factors that determine SHG membership and none at the time of this research have explored factors which determine the duration of membership once enrolled in SHGs.

The few studies [13–16] from India and Nigeria [17] that examined determinants of SHG membership found that membership is positively associated with increasing age for women, explained by greater social mobility afforded to women in traditional settings as they transition to roles of carers within families. While these studies offer guidance, the generalisation of these findings is limited due to small sample size (less than 300 women). Also, these studies included

a few economic determinants in the analyses. Moreover, studies from regional India have not adequately described sampling and statistical techniques involved [18–20].

Ensuring social programs and microfinance alike reach the intended populations is an important factor in determining program success. Understanding beneficiary characteristics is also important as SHGs are considered to be a scalable community-based model on which other socio-economic and health programs can be incorporated. SHGs, for example, in India are co-opted in village health promotion events such as the Village Health Nutrition Day (VHND) where preventive health services are delivered and monitored by SHGs [21]. Moreover, studies also show that for many poor populations such as SHG clients, out of pocket health expenses remain a major factor in over-indebtedness and in diverting loans away from enterprise or income-generating activities [4, 5, 22].

In India, the state of Uttar Pradesh (UP), the most populous, with high levels of poverty, low development and health indicators, presents the optimal setting for generating evidence on the reach of microfinance program and participating clients [23, 24]. This study examined the determinants of SHG membership and duration of membership once enrolled in integrated microfinance and health literacy program in rural Uttar Pradesh, India. The current study was built on past studies [14, 15, 25] and investigated a range of economic, socio-demographic and area-level characteristics that exhibit broader interconnectedness on membership and duration. The findings are intended to enhance our understanding of SHG programs in India and generate evidence to improve program design that may be more generalisable.

## Methods

### Ethics statement

Ethics approval was granted by the Human Research Ethics Committee, at the University of Canberra. Previously, ethics approval for the primary data collection under the IMFHL project was approved by the Independent Ethics Committee of Population Council, New York

This study used secondary data from an evaluation of an Integrated Microfinance and Health Literacy (IMFHL) program that was implemented from 2012 to 2017 in rural Uttar Pradesh (UP). The IMFHL program was a five-year initiative that sought to reach income poor women in villages across UP with a maternal and neonatal health literacy intervention that was integrated within a microfinance (SHG) program. The health intervention was designed to address community-related barriers in adopting preventive health behaviours to reduce maternal and neonatal mortality [25].

### Project context

The IMFHL program was funded by the Bill and Melinda Gates Foundation (BMGF) and involved a consortium of partners comprising of Public Health Foundation of India (PHFI), Population Council (PC), Boston University (BU), and Rajiv Gandhi Mahila Vikas Pariyojan (RGMVP).

RGMVP's SHG model was developed as a community-driven and scalable platform that aimed to empower marginalised and socially disadvantaged women in high poverty and low development regions of Uttar Pradesh [25]. Under the IMFHL program, implementation districts were selected comprising high maternal and neonatal mortality burden with a higher percentage of scheduled caste (SC)/tribe (ST) and low literacy [25]. Intervention and control blocks from the same districts were selected to receive either SHG (control blocks) or SHG plus health (intervention blocks) in the IMFHL program. RGMVP selected 120 intervention blocks based on their operational principles and separately identified 83 control blocks, matched as possible to the intervention blocks with respect to the percentage of SC/ST, where

only SHGs were established [25]. Women in villages were identified for SHG membership by RGVPM using community participatory approach and inclusion criteria that sought to identify the most disadvantaged households often represented by landless poor households with low literacy, lower social class (and caste) with multiple (social) deprivations [25]. In these villages, one member from each eligible household could join with other interested poor women in the same village to participate in SHG activities that were nurtured by RGMVP's field staff. Non-participating households were not part of RGMVP's SHG platform in program villages and would not be facing similar credit constraints as poorer member households.

As a first step, SHGs were established by RGMVP, and later the program consortium added a maternal and neonatal health literacy intervention. The health literacy component targeted SHG members and their households, primarily, pregnant mothers and mothers who had a baby in the last 12 months. These women primarily received health literacy through a SHG member trained in the health component that comprised of health messages and newborn care infection control and thermal care skills. Overall, the program used a Behaviour Change Communication (BCC) strategy to disseminate health literacy through various strategies and mediums such as inviting pregnant and new mothers to SHG meetings, targeted health letters to pregnant and new mothers, house visits and exposure to community health video shows developed by the program [25].

## Survey sampling approach and study population

**Sampling procedure.**    The data was collected from eligible women using a structured questionnaire in two cross-sectional pre- and post-intervention survey waves, 2015 and 2017, from 20 districts (out of a total of 75 districts in U.P state). U.P is administratively subdivided into 75 districts, 822 blocks and approximately 98,000 Gram panchayats (GP) with a total population of almost 200 million [26]. GPs are the smallest administrative units within blocks where SHGs are established. A GP is a larger main village and may have smaller villages attached to it. While the IMFHL program was implemented in GPs across 120 blocks in 33 districts, the survey data were collected from 70 blocks in 20 districts, and the sample is representative of the program's coverage area.

The surveys used a three-stage sampling approach for selecting blocks, gram panchayats (GPs), and finally, households based on the states' administrative hierarchy [26]. All the program intervention blocks were first arranged in ascending order with respect to and stratified by the percentage of scheduled caste or tribe, a critical parameter for development. Then the required number of intervention (SHG and health) blocks was equally selected by random sampling within each stratum [25]. Correspondingly, control (SHG only) blocks were selected within the same district and comprised of roughly similar proportions of SC/ST as intervention blocks. These matched control blocks were, however, selected independently of the intervention blocks. While the IMFHL program established SHGs in both intervention and control blocks, only households in intervention blocks received the additional health intervention. The survey design interviewed SHG member and non-member women from both control and intervention blocks. Gram Panchayats (GPs) were randomly selected to include diverse GP's in terms of population coverage by SHGs across three strata: 5–15 per cent, 16–30 per cent and 30–60 per cent. GP's with < 5 per cent and >60 per cent SHG population coverage were removed as outliers [25]. Within each selected GP, the survey collected data from the eligible woman in both program and non-program households following a systematic random sampling [25]. However, a mapping and house listing exercise in each selected village showed that the number of households with an eligible woman was almost equal to the sample size requirement [25]. Therefore, all program households with an eligible woman were selected for an

interview. As SHG programs enrol one member from each household only, each individual woman in the survey represents a household. If more than one eligible woman was found in a household, a random procedure was used to select the eligible respondent [25].

While the survey followed a stratified sampling strategy for blocks and Gram Panchayat selection within the blocks, GPs were not matched accordingly to the level of SC/ST as done for blocks. For the analysis in this paper, eligible women from both program and non-program households (from both the control and intervention blocks) were considered for the analysis of determinants of SGH membership and duration. In both survey rounds, data was collected from the same districts, blocks, and GPs, but *not* the same households or women. Moreover, data was also collected from households in the main village and (smaller) satellite villages connected to the main village and part of it. Furthermore, as the survey used different selection criteria at the higher level (stratified and matched block selection using SC/ST percentage) and at a lower level (stratified GP selection using population coverage by SHGs), this survey analyses all eligible women in households across all GPs. Thereby retaining the heterogeneity/ variation among households in different GP's that may be accounted for by different SHG membership and duration potential predictors.

**Sample size.** The survey comprised a total of 15,300 eligible women among whom 8,048 (53 per cent) were SHG members. Data were collected at the individual, household and village levels using separate questionnaires. The eligible respondents comprised currently married women aged 15 to 49 years who had babies in the 12 months preceding the survey; household head; and village representatives [25]. In order to capture the individual, households and community/area level of influence on SHG membership and duration, this analysis merged individual-level, household and village level sub-datasets across rounds.

## Statistical analysis

This study examined the determinants of SHG membership and duration of membership for women living in rural Uttar Pradesh. The explanatory variables included economic, socio-demographic, health and area-level characteristics. All analyses were performed using Stata 14 (Statacorp, USA).

The two dependent variables, SHG membership, and duration of membership, were defined as follows. Membership is a binary variable defined and coded as zero (No) for those eligible women that are non-member, and as one (yes) for an eligible woman who is *herself* a member of the SHG or *belongs* to a household where someone else (for example, mother, mother-in-law, sister in law) is a SHG member. The second outcome, duration of membership for SHG members *only*, is a count variable defined as the number of completed months of membership in the SHG once enrolled.

The explanatory variables used in this analysis are broadly supported by the program participation and maternal and child health literature and presented in the conceptual framework (Fig in S1 Appendix). The variables broadly represent three levels of potential influence, namely, i. economic, ii. socio-demographic and health and iii. area-level characteristics.

Economic variables included household poverty defined by having: i. Below Poverty Line (BPL) card, an objective income-based economic indicator of a household's poverty level determined by the Indian Government with eligible poor households provided with a BPL card [27, 28], ii. working status of the eligible woman for cash, kind or both, iii. a constructed wealth quintile using surveyed household's asset ownership and iv. loan purpose. The extent of poverty is not homogenous among the poor with levels or gradations of poverty, extending from the marginally poor to the "poorest" often described to be sitting at the bottom of the poverty pyramid. Principal component analysis (PCA) was used in this paper to construct the

wealth score to draw out poverty gradients using household assets and amenities variables. Wealth scores were then categorised into quintile groups comprising an equal proportion of households (20 per cent) and extending from the marginally poor in the first quintile down to the poorest in the fifth quintile. The fourth economic variable loan purpose is defined as any loan taken amounting to more than USD seven (Rs 500) in the last three years from any source. Sources of borrowing for Non-SHG members include family/friends/banks/ informal money lenders, but not through SHGs. The analysis included the variable 'round' that represents either survey round I (2015) or II (2017) and also includes the interaction of quintile with loan purpose to examine variations in borrowing purposes amongst gradients of the poor that may have different associations with membership and duration.

The analysis included additional socio-demographic and health variables namely: family type (nuclear versus. joint), religion (Muslim, Hinduism and Others), household social caste (General Caste, Other Backward Caste, Scheduled Tribes-protected indigenous population, and Scheduled Caste-a subdivision of the Hindu Caste system which is also a proxy for social marginalisation), education level of the eligible woman and her husband, parity-number of completed pregnancies, place of last delivery, pregnancy loss and the number of contacts with a health worker during the last pregnancy. Area-level variables included: availability of government or private health facility in the village, number of private doctor clinics or pharmacies in a village, number of community health workers in a village.

Separate multivariable regression analyses were conducted for each of the two outcomes. Firstly, to assess the determinants of SHG membership, logit regression models were performed, where explanatory variables were included in a forward selection of stepwise regression. The SHG membership logit models assessed the impact of explanatory economic variables in the first model, followed by the inclusion of socio-demographic and health variables in the second model and finally, the area-level variables in the third model. For conciseness, only results from the final multivariable logistic regression (Model III) are presented in Table 2; however, results of all logit models I-II-III are provided in Table in the S4 Appendix.

Secondly, the analyses also investigated the determinants of duration of SHG membership, using a negative binomial hurdle regression model which accommodated for the two-step decisions involved in microfinance membership: first, the decision to join a microfinance group, and then the decision to continue membership contingent on joining. More specifically, the model allowed for the decision to join a SHG group to be treated independently from the decision to continue membership (duration) in *one integrated model*. The negative binomial hurdle model is appropriate for count outcomes with underlying data overdispersion, as observed in this sample [29–32]. The hurdle addresses the excess zeroes found in this data representing non-members as well as members with less than one-month membership duration, both coded as zero (duration = 0).

Functionally, the hurdle partitions the model into two processes. First, a binary process that generated positive counts for SHG members ('1') versus non-member ('0') is fitted using a Logit model [29]. Second, the model fitted the positive counts only, that is SHG membership duration (number of months continued as a member) using a negative binomial regression model. The estimated produced a two-component regression result: first, the probability of being a member (1) or non-member (0) using 'Odds ratios (OR), and second, a count component that predicted the duration of the membership using Incidence Rate Ratio (IRR) [29]. However, we reported the IRR only from the full regression model in this paper, that included all three groups of explanatory variables in the analysis.

Summary statistics on the duration of membership revealed that the sample variance [31] is much greater than the mean value [21], which is an indication of over-dispersion in the data. In such instances, the negative binomial regression procedure, which is designed to handle

**Table 1. Summary statistics of selected variables by Non-Self-Help Group [Non-SHG] Household and Self-Help Group [SHG] households.**

| Sr. no | Variable | Summary Statistics(N = 15,300) | |
|---|---|---|---|
| | | Non-SHG Households (= 0) Mean/ Proportion (SD) | SHG Households (= 1) Mean/ Proportion (SD) |
| | *Dependent Variable* | | |
| 1. | SHG Membership status (= 1 if the eligible woman is herself a member or belongs to an SHG member household) | 47% | 53% |
| 2. | SHG Duration (in completed months once enrolled into a SHG) | NA | 42 (33) |
| | *Independent Variables: Economic Characteristics* | | |
| 3. | **Household (HH) with Below Poverty Line (BPL) Card** | 40% | 51% |
| 4. | **Household Wealth Quintile (Poor to Poorest)** | | |
| | 1. Marginally Poor | 22% | 19% |
| | 2. Moderately Poor | 19% | 20% |
| | 3. Poor | 20% | 21% |
| | 4. Poorer | 20% | 20% |
| | 5. Poorest | 19% | 20% |
| 5. | **Purpose of Last Loan Borrowed by SHG and Non-SHG Members from any source.** | | |
| | 1. No Loan Taken | 72% | 61% |
| | 2. Enterprise Purposes (To start or expand the business) | 3% | 10% |
| | 3. Non-Enterprise(Repay old loan, Child's education or Marriage, House Repairs) | 6% | 11% |
| | 4. Health Purpose(For Treatment of Illness or Delivery) | 5% | 12% |
| | 5. Others(reason not stated) | 4% | 6% |
| | *Independent Variables: Socio-demographic and Health aspects* | | |
| 6. | **Eligible Woman Presently Working to Earn in Cash, Kind or Both** | 13% | 17% |
| 7. | **Scheduled Caste** | | |
| | General Caste | 14% | 11% |
| | Other Backward Caste | 44% | 40% |
| | Scheduled Tribe | 6% | 5% |
| | Scheduled Caste | 36% | 44% |
| 8. | **Eligible Woman(EW's) Education Level** | | |
| | No schooling | 34% | 33% |
| | Completed Primary& Middle School(year nine) | 38% | 41% |
| | Completed Secondary(up to year 10) and Above | 28% | 26% |
| 9. | **EW's Husband Education Level** | | |
| | No schooling | 17% | 17% |
| | Completed Primary& Middle School(year nine) | 46% | 51% |
| | Completed Secondary(year 10) and Above | 37% | 32% |
| 10. | **Parity of Eligible Woman** | 2.3 (1.4) | 2.5 (1.5) |
| 11 | **Total Pregnancy Loss** | 0.35 (0.76) | 0.41 (0.83) |
| 12. | **Place of Last Delivery** | | |
| | Home Delivery | 12% | 9% |
| | Institutional Delivery | 88% | 91% |
| | *Independent Variables: Area (Village) Level Characteristics* | | |
| 13. | **Mean Number of ASHA & ANM in village** | 1.8 (2.1) | 2.0 (2.4) |
| 14. | **Mean number of contact with ASHA/ANM/AWW/SHG in last pregnancy** | 3.7 (5.3) | 4.5 (5.7) |
| | *Independent Variables: Round Characteristics* | | |
| 15. | **Evaluation Survey Round** | | |
| | Round 1 /Baseline-2015 (= 0) | 61% | 51% |

(*Continued*)

**Table 1.** (Continued)

| Sr. no | Variable | Summary Statistics(N = 15,300) | |
| --- | --- | --- | --- |
| | | Non-SHG Households (= 0) Mean/ Proportion (SD) | SHG Households (= 1) Mean/ Proportion (SD) |
| | Round 2/ Endline-2017 (= 1) | 9% | 49% |

**Acronyms:** Eligible woman (EW), Accredited Social Health Worker (ASHA)/Auxiliary Nurse Midwife (ANM) Anganwadi worker (AWW)-ASHA/ANM & AWW are government health workers situated in villages as per population guidelines and provide preventative maternal, child and other health services.

such extra Poisson variations, would be more appropriate as it produces more valid estimates and best model fit [29]. Also, out of 8,048 SHG members in our sample, 99.5 per cent or 8,003 were members for a duration of more than one month. Moreover, only 0.5 per cent of SHG members or 45 respondents were members for less than one month also coded as zero. Differentiating these zeroes (for members with less than one-month membership) from non-members also coded as zeroes are possible through the hurdle process described above. The suitability of the hurdle negative binomial (HNB) model against other count data models with the lowest Consistent Akaike Information Criterion (CAIC) and Bayesian Information Criterion (BIC) statistic is presented in the Table in the S2 Appendix [29]. The regression results from the hurdle negative binomial model analysis for SHG duration are also presented in Table 2 (Column 2).

## Summary statistics

Table 1 shows summary statistics for selected variables showing key similarities and differences between the Non-Self Help (SHG) Group member households and Self-Help Group (SHG) member households. Summary statistics for *all* variables by Non-SHG and SHG groups is also provided in the full version of Table in the S3 Appendix.

The statistics revealed that 51 per cent of SHG households had a Below Poverty line (BPL) card (an income-based limit to identify poverty) while 40 per cent of non-members held a BPL card. The descriptive statistics also revealed comparable characteristics across SHG and non-SHG households with respect to woman's age (mean 26 years), woman education (34 per cent had no schooling), and maternal health indicators. Eligible women across both groups (SHG & Non-SHG) reported a mean parity close to 2.3 reflecting near current Indian fertility rates (median 2.2, range 2.1–4) with close to 90 per cent reporting institutional deliveries [34]. SHG members had slightly more contacts (4.5) with frontline health workers during the last pregnancy as compared to non-SHG members (3.7). The loan purpose (purpose of the last loan) statistics revealed that overall more SHG member households as compared to non-members borrowed a loan in the last three years' time which was more than USD seven (Indian Rupees 500). Moreover, a greater per cent of SHG households compared to non-SHG households borrowed their last loan either for purposes relating to enterprise (to start/expand a business) or for non-enterprise (repay an old loan, child's education or Marriage, house Repairs). However, an almost similar number of non-SHG (15%) households as compared to SHG (13%) members borrowed their last loans to meet health-related expenses.

## Results

### Results for determinants of SHG membership

Table 2 presents results from the full logit regression model (Model III) for SHG membership using the complete set of explanatory variables, that is, economic, socio-demographic, health

**Table 2. Results from the multivariate logit regression for determinants of SHG membership and hurdle negative binomial regression for determinants of SHG duration of membership.**

| | Variables | SHG Membership: Logit Analysis | SHG Duration: Hurdle Negative Binomial Analysis |
|---|---|---|---|
| | | OR (95% CI) | IRR (95% CI) |
| Sl. No | *Economic Variables* | | |
| 1. | **Household (HH) has Below Poverty (BLP) Card** | | |
| | No(reference) | | |
| | Yes | 1.87*** (1.57–2.23) | 0.99 (0.91–1.07) |
| 2. | **Quintile (rich to poor)** | | |
| | 1. Marginally Poor (reference) | | |
| | 2. Moderately Poor | 1.42*** (1.21–1.67) | 0.93* (0.85–1.01) |
| | 3. Poor | 1.35*** (1.14–1.59) | 0.86*** (0.79–0.93) |
| | 4. Poorer | 1.65*** (1.39–1.94) | 0.81*** (0.74–0.89) |
| | 5. Poorest | 1.71*** (1.45–2.02) | 0.78*** (0.71–0.85) |
| 3. | **HH has Below Poverty Card #Quintile (Rich to Poor)** | | |
| | BPL card (Yes)#Quintile 1 (Marginally Poor)(reference) | | |
| | BPL card (Yes)#Quintile 2 (Moderately Poor) | 0.92 (0.74–1.13) | 1.07 (0.96–1.17) |
| | BPL card (Yes)#Quintile 3 (Poor) | 0.94 (0.76–1.16) | 1.05 (0.95–1.16) |
| | BPL card (Yes)#Quintile 4 (Poorer) | 0.80** (0.65–1.00) | 1.04 (0.94–1.15) |
| | BPL(Yes)#Quintile 5 (Poorest) | 0.89 (0.71–1.11) | 1.04 (0.94–1.15) |
| 4. | **Eligible Woman Presently Working status** | | |
| | Not Working (reference) | | |
| | Presently Working to earn in cash, kind or both. | 1.25*** (1.13–1.38) | 0.99 (0.94–1.03) |
| 5. | **Last Loan Purpose** | | |
| | No Loan availed(reference) | | |
| | Enterprise reasons | 4.52*** (3.08–6.63) | 1.73*** (1.52–1.97) |
| | Non- Enterprise reasons | 3.10*** (2.24–4.28) | 1.78*** (1.56–2.03) |
| | Health and Illness | 1.71*** (1.28–2.27) | 1.38*** (1.20–1.60) |
| | Others(Reason not stated) | 2.28*** (1.43–3.63) | 1.42*** (1.18–1.71) |
| 6. | **Evaluation Round** | | |
| | Round1(reference) | | |
| | Round 2 | 1.63*** (1.48–1.78) | 2.69*** (2.56–2.82) |
| 7. | **Round #Last Loan Purpose** | | |
| | Round 1#No Loan Taken (reference) | | |
| | Round 2#Loan taken for Enterprise reasons | 0.42*** (0.30–0.60) | 0.55*** (0.50–0.61) |
| | Round 2#Loan taken for Non–Enterprise reasons | 0.72*** (0.57–0.94) | 0.59*** (0.54–0.66) |
| | Round 2#Loan taken for Health and Illness reasons | 0.62*** (0.50–0.76) | 0.61*** (0.53–0.70) |
| | Round 2#Loans taken for Other purposes | 0.48*** (0.34–0.69) | 0.61*** (0.53–0.69) |
| 8. | **Last Loan Purpose #Quintile(Rich to Poor)** | | |
| | No Loan Taken #Quintile 1 (reference) | | |
| | Loan for Enterprise Reasons# Quintile 2(Moderately Poor) | 1.31 (0.82–2.08) | 1.09 (0.95–1.26) |
| | Loan for Non-Enterprise Reasons#Quintile2(Moderately Poor) | 0.64*** (0.44–0.92) | 0.98 (0.86–1.12) |
| | Loan for Health Reasons# Quintile 2(Moderately Poor) | 0.56*** (0.40–0.79) | 1.18** (1.01–1.37) |
| | Loan for Other Reasons# Quintile 2(Moderately Poor) | 1.18 (0.71–1.97) | 1.02 (0.86–1.22) |
| | Loan for Enterprise Reasons# Quintile 3(Poor) | 1.31 (0.83–2.08) | 1.09 (0.94–1.26) |
| | Loan for Non-Enterprise Reasons #Quintile3(Poor) | 0.61*** (0.42–0.87) | 1.11 (0.97–1.26) |
| | Loan for Health Reasons # Quintile 3(Poor) | 0.66*** (0.47–0.91) | 1.13* (0.98–1.31) |
| | Loan for Other Reasons# Quintile 3(Poor) | 1.35 (0.82–2.22) | 1.09 (0.93–1.29) |
| | Loan for Enterprise Reasons# Quintile 4(Poorer) | 1.69** (1.00–2.85) | 0.86** (0.75–0.99) |

*(Continued)*

**Table 2.** (*Continued*)

| | Variables | SHG Membership: Logit Analysis | SHG Duration: Hurdle Negative Binomial Analysis |
|---|---|---|---|
| | | OR (95% CI) | IRR (95% CI) |
| | Loan for Non-Enterprise Reasons #Quintile 4(Poorer) | 0.75 (0.49–1.13) | 1.06 (0.91–1.23) |
| | Loan for Health Reasons # Quintile 4(Poorer) | 0.55*** (0.39–0.76) | 1.15 (0.99–1.33) |
| | Loan for Other Reasons# Quintile 4(Poorer) | 0.64* (0.39–1.04) | 1.12 (0.42–1.34) |
| | Loan for Enterprise Reasons# Quintile 5(Poorest) | 1.52 (0.86–2.69) | 0.89* (0.76–1.04) |
| | Loan for Non-Enterprise Reasons #Quintile 5(Poorest) | 0.69* (0.45–1.04) | 0.98 (0.85–1.14) |
| | Loan for Health Reasons# Quintile 5(Poorest) | 0.44*** (0.31–0.61) | 1.11 (0.96–1.30) |
| | Loan for Other Reasons# Quintile 5(Poorest) | 0.64* (0.38–1.07) | 1.02 (0.85–1.22) |
| | *Socio-demographic Variables* | | |
| 9. | **Type of Family** | | |
| | Nuclear Family (reference) | | |
| | Joint & Extended Family | 1.33*** (1.23–1.44) | 1.18*** (1.13–1.22) |
| 10. | **Religion (Household Head)** | | |
| | Muslim Household(reference) | | |
| | Hinduism & Others | 1.19*** (1.05–1.35) | 0.96 (0.90–1.02) |
| 11. | **Caste** | | |
| | General Caste(reference) | | |
| | Other Backward Caste | 0.95 (0.85–1.07) | 0.93*** (0.88–0.99) |
| | Scheduled Tribe | 1.04 (0.87–1.24) | 0.88*** (0.81–0.97) |
| | Scheduled Caste | 1.25*** (1.12–1.41) | 0.86*** (0.81–0.91) |
| 12. | **Eligible woman (EW) Age in years** | 1.01*** (1.00–1.02) | 1.02*** (1.01–1.02) |
| 13. | **Eligible woman education level(completed years)** | | |
| | No schooling(reference) | | |
| | Completed Primary& Middle School (up to year 9) | 1.19*** (1.09–1.30) | 0.97 (0.93–1.02) |
| | Completed Secondary(up to year 10) and Above | 1.13** (1.01–1.26) | 0.91*** (0.86–0.96) |
| 14. | **EW's Husband Education level(completed years)** | | |
| | No schooling(reference) | | |
| | Completed Primary& Middle School(up to year 9) | 1.09* (0.98–1.19) | 1.07*** (1.02–1.11) |
| | Completed Secondary(up to year 10) and Above | 0.93 (0.83–1.05) | 1.13*** (1.07–1.19) |
| 15. | **Parity** | 1.10*** (1.06–1.15) | 0.98* (0.96–1.00) |
| 16. | **HH Below poverty Card # Parity** | | |
| | No BPL card#Parity(reference) | | |
| | Yes card# Parity | 0.89*** (0.85–0.94) | 1.01 (0.99–1.03) |
| 17. | **Total Pregnancy Loss** | 1.05** (1.00–1.09) | 1.02** (1.00–1.04) |
| 18. | **Place of Last Delivery** | | |
| | Home Delivery(reference) | | |
| | Institutional Delivery | 1.10 (0.98–1.23) | 1.09** (1.01–1.17) |
| | *Area Level Variables* | | |
| 19. | **Availability of Health facility in the village** | | |
| | No Health Facility(reference) | | |
| | Only Government Health Facility | 0.95 (0.88–1.03) | 1.03** (0.99–1.07) |
| | Only Private Health Facility | 0.94 (0.71–1.26) | 1.03 (0.92–1.16) |
| | Both Government & Private Health Facility | 0.83 (0.83–1.11) | 1.06** (1.00–1.13) |
| 20. | **Total Number of Private Doctor Clinics in village** | 1.02 (0.97–1.07) | 0.99 (0.98–1.01) |
| 21. | **Number of community health workers(ASHA & ANM) in the village** | 1.06*** (1.04–1.08) | 0.98*** (0.98–0.99) |
| 22. | **Number of contact with ASHA/ANM/AWW/ SHG in last pregnancy** | 1.01*** (1.01–1.02) | 1.00 (1.00–1.00) |

(*Continued*)

**Table 2.** (Continued)

| Variables | SHG Membership: Logit Analysis | | SHG Duration: Hurdle Negative Binomial Analysis | |
|---|---|---|---|---|
| | OR (95% CI) | | IRR (95% CI) | |
| **Estimation of Model Fit** | SHG Membership Logit Analysis | | SHG Duration (HNB) | |
| Log likelihood | -9,945 | | -45,713 | |
| Number of Observation | 15,300 | | 15,300 | |
| AIC/BIC | 20,002 | 20,430 | 91,653 | 92,516 |

Confidence intervals in parentheses; and significant p-value showing 0.01***,0.05** and 0.10* levels. Log-likelihood and AIC/BIC values were also reported.

system and area-level variables. The model was estimated using the maximum likelihood estimation method. The Odds Ratios (OR) are reported as measures of the strength of association, along with their associated 95% confidence intervals (95%CI) and p values. Details of the model III are presented as follows.

**Economic, socio-demographic and area-level determinants of SHG membership.** The results from economic variables showed that the odds of membership are 1.87 times higher for poor households with a Below Poverty line Card (BPL) than households without BPL cards or economically better households (OR 1.87, 95%CI:1.57–2.23, p<0.01). The wealth quintile variable estimated in this study using the surveyed households asset data revealed that compared to the base category of marginally poor households, the overall odds of becoming a member *increased* as the wealth quintiles *descended* from marginally poor households (quintile one) to the poorest (quintile five) in this sample suggesting that poorer households are more likely to be captured for membership by the program. The poorest households are in fact, almost 70 per cent or 1.71 times *more likely* to be members as compared to marginally poor households in the same villages (OR = 1.71, 95%CI:1.45–2.02, p<0.01). However, results from the interaction of BPL card households (HH) with wealth quintiles, showed that for those households with a BPL poverty card, their likelihood of membership *decreased* when the wealth quintiles *descended (*marginally poor to poorest) suggesting that BPL households are less likely to be SHG members as they become poorer. However, results only for the poorer households (quintile four) with BPL cards were significant in the final model III (OR = 0.80, 95%CI:0.65–1.00, p<0.05).

For another socio-economic indicator, working status, results showed that eligible women presently working were 25 per cent more likely to be members than non-working women (OR = 1.25, 95%CI:1.13–1.38, p<0.01). Results pertaining to the inclusion of socio-demographic variables revealed higher membership likelihood for joint/extended families as compared to nuclear families (OR = 1.33, 95%CI:1.23–1.44, p<0.01)

Results of the religion and social class structure revealed a higher likelihood of becoming a member for women belonging to Hindu religion compared to Muslim households (OR = 1.19, 95% CI:1.05–1.35, p<0.01). Additionally, caste structure revealed that women belonging to Scheduled Caste, a proxy for social marginalisation are 25 per cent more likely to be members compared to women belonging to general caste households (OR = 1.25, 95%CI:1.12–1.41, p<0.01).

The impact of the level of education on the likelihood of becoming a member was a positive association showing women who completed primary education, almost 19 per cent more likely to be a SHG member or belong to SHG household as compared to a woman with no schooling

(OR = 1.19, 95%CI:1.09–1.30, p<0.01). However, the level of the woman's husband's education came up with a negative but non-significant association with membership.

The results from membership associations with health and fertility indicators revealed that in general, the likelihood of membership increases with increase in parity, that is, the number of pregnancies carried by women to gestation age (OR = 1.10, 95%CI:1.06–1.15, p<0.01), and also increases when the total number of pregnancy loss rises (OR = 1.05, 95%CI:1.00–1.09, p<0.01). Interestingly, the results revealed that as households with Below Poverty Line (BPL) cards experience increasing parity, their likelihood of SHG membership reduces (OR = 0.89, 95%CI:0.85–0.94, p<0.01).

Inclusion of area-level variables revealed the likelihood of SHG membership increased for women in villages with a higher number of community health workers (OR = 1.06, 95% CI:1.04–1.08, p<0.01), and also increased with an increasing number of contacts with a community health worker (ASHA/AWW/ANM) or SHG member during their last pregnancy (OR = 1.01, 95%CI:1.01–1.02, p<0.01).

Moreover, the model also estimated the effects of loan purpose, a variable, that is related to the household's last reason for borrowing money in the last three years, and the interaction effects of loan purpose by wealth quintiles. Overall, the results showed that the likelihood of membership is higher when households borrowed loans for any reason that includes enterprise/non-enterprise or consumption/health and illness. Within the loan purpose categories the likelihood of becoming a member is highest when they borrowed for enterprise purposes, for example, to start or expand business (OR = 4.52, 95% CI:3.08–6.63, p<0.01) and the lowest when they borrowed for a *health purpose*, for example, to meet the cost for illness, delivery expense (OR = 1.71, 95%CI:1.28–2.27, p<0.01) compared to households that did not borrow.

Additionally, to determine the effect of IMFHL project intervention between the two survey rounds, the model also estimated the effect of round II compared to round I, and the round by loan purpose interaction effects. The results revealed that while households in round II are 1.63 times more likely to be SHG members(OR = 1.63, 95%CI:1.48–1.78, p<0.01), the likelihood of membership decreased for all-round II households when they borrowed for *any reason*. In round II the households that borrowed for enterprise reasons were 0.42 times less likely (OR = 0.42, 95% CI:0.30–0.60, p<0.01), and that borrowed for health reasons, almost 0.62 times less likely (OR = 0.62, 95%CI:0.50–0.76, p<0.01), to be members as compared to households that were in round II but, did not borrow any loans.

The link between access to finances, poverty and health was observed from the interaction of wealth quintile with loan purpose. The estimated interaction effects showed that on the whole, borrowing loans for health expenses are negatively associated with the likelihood of being a member for all quintiles of the poor, with the poorest households being 0.44 times *less likely* to be members as compared to the marginally poor (quintile one) who did not take any loans (OR = 0.44, 95%CI:0.31–0.61, p<0.01).

## Results for determinants of duration of membership in SHGs

The results from the hurdle negative binomial (HNB) regression model are presented in Table 2 (column 2) with the Incidence Rate Ratio (IRR) values reported along with their associated 95% confidence intervals (95%CI). The explanatory variables used in this model are the same as those used in the logit regression model for the SHG membership analysis.

**HNB regression results for duration of membership in SHG.** The results showed that once enrolled into SHGs, women belonging to households with below poverty line (BPL) cards were less likely to remain, members, as compared to households without BPL cards (IRR = 0.99, 95%CI:0.91–1.07, p >0.05). The likelihood of staying members (duration)

*decreased* across the poverty gradient represented by the wealth quintile with the poorest (quintile five) households being 0.78 times *less* likely to stay members as compared to the poor (IRR = 0.78, 95%CI:0.71–0.85, p<0.01)

Like the SHG membership model, this duration model also revealed that joint families as compared to nuclear families are more likely to continue as SHG members for longer (IRR = 1.18, 95%CI:1.13–1.22, p<0.01). Conversely, poor households belonging to disadvantaged social class (scheduled castes), compared to general caste, are less likely to stay members (IRR = 0.86, 95%CI:0.81–0.91, p <0.01).

The levels of education attained by women and their husbands have opposite and significant effects on duration. While women with secondary education and above are **less** likely to continue membership (IRR = 0.91, 95%CI:0.86–0.96, p<0.01) as compared to women with no schooling, eligible women whose husbands completed secondary education or above are more likely to continue membership for longer compared to women whose husband has had no schooling (IRR = 1.13, 95%CI:1.07–1.19, p<0.01).

Results from other socio-demographic and health variables revealed that while women who reported higher pregnancy loss were more likely to stay members (IRR = 1.02, 95%CI:1.00–1.04, p<0.05), women with an increasing parity were *less* likely to stay members; however, this association is only significant at a 10 per cent level of significance (IRR = 0.98, 95%CI:0.96–1.00, p = 0.10). The model also pointed out the significant effects of area-level factors on SHG duration. The findings revealed that women in villages where both government and private health facilities were available were more likely to continue membership for a longer period (IRR = 1.06, 95%CI:1.00–1.13, p<0.05).

As the second round of survey data collection was completed after two years of the IMFHL program implementation in 2017, the model found that households interviewed in round II were 2.69 times more likely to continue as members compared to those interviewed in the round I (2015) when program implementation (health layering) was yet to begin (IRR = 2.69, 95%CI:2.56–2.82, p<0.01). The results also revealed that households that borrowed loans for *any* purpose were more likely to stay members for longer. Indeed, households that borrowed for enterprise purposes were more likely to continue as members (IRR = 1.73, 95% CI:1.52–1.97, p<0.01), compared to households that did not borrow any loans.

Moreover, similar to the association observed between SHG membership and both survey round and interaction of round with loan purpose, the model for SHG duration revealed that households in round II are more likely to continue as members, but households that borrowed loans (for any purpose) in round II were less likely to continue their SHG membership for longer.

The SHG duration model also included the interaction of wealth quintile with loan purpose (quintile by loan purpose) to examine the effects on duration in SHGs among gradients of poor and the different reasons for which they borrowed money from any source (bank, moneylender, and family/friends/neighbour). The interaction results revealed that poorer households, in fourth and fifth quintiles, were *less likely* to remain SHG members when they borrowed loans for enterprise purposes as compared to poor households (quintile one) that did not borrow any loans (IRR = 0.86, 95%CI:0.75–0.99, p<0.05).

## Discussion

This study makes a significant contribution to the literature in investigating the determinants of membership and duration of membership of an integrated health literacy and microfinance based SHG program in India. Another study [15] from India looked at how microfinance only SHGs evolved over time using a survival analysis model and found higher rates of drop-outs

from SHG (up to 20 per cent) among disadvantaged castes and households without landholdings. That study [15] also, found a protective effect of some formal education on the duration of the membership. Our analysis used a sophisticated and modified count data model to examine the factors that determine how long a woman/household remained as SHG member when they are already enrolled as SHG members in Uttar Pradesh. Moreover, by analysing determinants of membership and duration of membership in one integrated model, this study provides evidence to guide SHGs programs to improve their implementation designs to retain clients. Furthermore, this study has significant policy implications as recent research on microfinance effectiveness recommend programs should support poor households for longer duration till they move up to the next rung of poverty [22]. The focus on the duration of membership, therefore, echoes a commitment to assist SHGs in identifying the right 'poor' clientele for membership and then retain them as members till they cross the poverty threshold at a minimum. Moreover, the study also improves on previous study designs by using a range of explanatory variables that explains much of the variation in the outcome variables.

More importantly, the study shows that poor households with Below Poverty Line cards are more likely to be SHG members as compared to poor households without BPL cards. Encouragingly, the study also finds that all gradients of the poor as captured through the wealth quintile constructed in this study using household assets are more likely to be members. Importantly, the poorest households based on the wealth quintile (quintiles four and five) are most likely to be SHG members. The findings from the wealth quintiles contrast previous evidence from regional India that showed exclusion of the poorest of the poor in typical microfinance programs [15, 16, 18–20, 33, 34]. Programs such as the specialised SHG plus health literacy model evaluated in this research that primarily seeks to target the poor and poorest are adequately able to capture the poor as found through the BPL statistics. However, the use of an additional measure such as the wealth quintile, as demonstrated in this study, will bring in much more required insight for identifying the gradients of poverty even among the BPL cardholders.

These findings essentially highlight potential gaps that other programs using similar (SHG) model may have been exposed to when the poorest households are identified through income-based economic variables such as using the BPL card alone in contrast to wealth-based quintiles. The research also adds new evidence on what factors influence continued membership or 'duration'. Our study found that the likelihood of continuing as members in SHG (duration) decreases for all gradients of poor households when they are captured through the wealth quintile which is contrary to the desired program goals [25]. SHG programs such as in this study sought to capture and then retain marginalised households for the duration of the program so that continued program support would enable households to build skills and accumulate resources and thereby, graduate to the next level of poverty. SHG program(s), are built on the premise of the poor self-managing and sustaining microfinance groups to maintain access to credit sources and decrease reliance on NGO's and external agencies. That will, in turn, help in reducing program implementation costs. The results from the SHG duration analysis show that the likelihood of continuing membership decreases as households gets poorer with the poorest households, as captured through the fifth wealth quintile almost 22 per cent or 0.78 times less likely to continue as members as compared to relatively better-off or marginally poor households. This suggests that all gradients of the poor, especially the poorest, potentially gain from being members when provided membership opportunity; however, they are more likely to discontinue from SHG membership potentially missing out from much-needed credit and welfare entitlements.

This study revealed that poor households with BPL cards are 87 per cent more likely to join SHGs demonstrating the programs ability to capture these poor households. BPL is an income

limit threshold used as a benchmark by the Indian Government to identify poor households eligible for entitlements when the annual income is less than USD 376 (Rs 26,000) per annum [35]. BPL is used as a benchmark for poverty estimation as it is a convenient measure for broad population-level screening, and that also makes interpretation and comparisons sub-nationally across regions and elsewhere relatively easier. Despite BPL being the preferred economic measure, the use of BPL cards as an effective criterion to capture the poor is widely debated.

Challenges associated with BPL have been highlighted [27, 28], and an Indian Government report [36] found that up to half of all eligible poor households from rural India were missed due to BPL targeting issues in capturing the poor. Some studies found that richer households exploited targeting errors in the BPL inclusion criteria to get a BPL card and be eligible to receive associated entitlements. Limitations are also recognised by Indian federal poverty alleviation program, the National Rural Livelihood Mission that uses a Participatory Inclusion Process to identify clientele instead of using BPL cards [37]. Similar inferences against the use of BPL cards as an instrument for identifying poor can be applied to the context of Uttar Pradesh state, and the program population analysed in this paper. Therefore, this paper adds significant value to the literature on potential gap in targeting the poor and poorest by poverty alleviation programs in India by constructing and using a wealth quintile to capture poverty and its gradients beyond BPL status. Other regional studies [15, 16, 18–20, 33, 34] from India have also, previously highlighted the limitations of microfinance programs in effectively reaching the poorest. These studies have constructed wealth quintiles based on households' land holding, sources of income and amount earned to identify the poorest, bypassing the use of formal measures of BPL.

## Targeting the poorest from the lowest wealth quintile households for SHG membership and duration

Previous studies showed that traditional microfinance models mostly served poor clients that were at or just below the poverty line [2, 10, 15]. A specialised type of microfinance, SHGs, in contrast, are considered better suited in reaching the poorest *within* the BPL population, as SHGs inherently operate on a pro-poor developmental agenda [10]. However, studies elsewhere showed that the poorest can still be excluded from some SHGs due to reasons related to caste-based social marginalisation, lack of minimum household finances to join groups, and intrinsic biases of implementing organisations preferring to enrol better off populations that are more easily reached [3, 10, 15]. SHGs by design focus on inclusion of poor clients at the *start* of programs with the assumption that once enrolled, the selected clients stay members. Contrary to other studies [15, 16], this paper points out that the poorest when identified through the wealth quintile (but not through the BPL card), are more likely to be SHG members; however, the analysis generates new evidence for the state of UP showing that once poorer households joined SHGs, they are unlikely to stay members for longer. While the poorest households represented by the last or fifth wealth quintile in this study are almost 70 per cent *more likely* to be SHG members at the start, but, after joining, they are 22 per cent *less likely* to continue as members as compared to better-off households. This suggests that the benefit of joining SHGs may be evident for poorer households at the time of joining SHGs, meaning that the program is better able to capture poorer households for SHG membership. However, the duration analysis revealed the reduced likelihood of continuing as members for all poor categories, especially for the poorest households as captured through the wealth quintile. This suggests that the program membership is not conducive for poorer households to stay SHG members; consequently, they are deprived from obtaining the full benefit of higher

loan eligibility and group saving that builds over time. This conclusion is also reached in this paper through the examination of membership and its duration using the Below Poverty Line (BPL) variable, which shows that poorer households with a BPL card are more likely to be SHG members but not likely to keep their membership longer as compared to poor households without BPL cards.

The wealth quintiles constructed in this study were proved to be potentially better at capturing/representing the poor in rural India compared to the BPL card status since the former involves multiple household assets ownership and access to facilities. The results from the interaction of BPL and quintiles showed that the likelihood of being a SHG member (membership) decreases as we move down the wealth quintiles of poor households that have a BPL card as compared to households without BPL cards. This may be due to BPL cards failing to reach the poorer households which are better captured by the wealth quintiles. The use of broader criteria such as the wealth quintiles is supported in the literature that recommends identifying the poorest using multidimensional indicators of deprivation that are better able to capture the varied manifestations of poverty [27, 28].

## Effect of borrowing on SHG membership and duration

Microfinance is based on the premise that poor households can improve their social outcomes when they are provided accessible and affordable financial services for investing in income generation activities, but also for managing economic shocks and repaying loans at low-interest rates [3, 4, 9, 10]. However, pragmatic views on microfinance highlight that not all poor borrowers could generate profit from loans as not everyone has the necessary entrepreneurial skillset to turn adversity into profit. Moreover, limited enterprise opportunities in rural markets influence economic opportunity in the first place for poor households [2, 6]. Accurate estimation of how much and whether households gain from membership is limited without additional time-series data on household's movement out of poverty. The current study instead used an explanatory proxy variable (loan purpose), to study its effects on membership and duration looking at the reasons for which households last borrowed more than USD seven (Rs 500). Four categories (Enterprise/Non-enterprise/Health/Others-reason not stated) are created to classify the reasons for borrowing the last loan both by SHG members or Non-member HH from any source. The results showed that in general when women borrowed funds for any purpose, they were more likely to become SHG members and were also more likely to continue membership for a longer period as was originally envisaged by microfinance advocates. A household borrowing for enterprise reasons for example is 4.52 times more likely to become a SHG member, but then, their membership duration on average is about 1.73 times more than the average duration of non-borrowing households. Encouragingly, households that borrowed funds for health reasons, such as delivery expenses or treatment for illness, were 1.71 times more likely to become SHG members and then, stay on average 1.38 times longer as a member, which may reflect their ability to gain from SHG membership and be able to pay back loans. While SHGs provide loans that can primarily be used for poverty reduction through investing in income-generating activities, loans borrowed for health expenses are considered to potentially divert funds from enterprise opportunity and keep households in the cycle of continuing borrowing for illness and associated debts. The burden of borrowing for health reasons in this sample is also shown in Table 1 as being the most common reason why SHGs members (12 percent) and non-members (15 percent) borrowed last time. The indicator variable representing borrowing for health reason, exerts similar positive effects on SHG membership and duration. This suggests the awareness among rural households of different purposes for which credit/loan can be availed from SHG's. This may also indicate that SHGs are

supporting general poor households with access to health funds during illness, unlike a formal bank that do not have provision for emergency health loans to income-poor families apart from public financed health insurance schemes. However, even though a loan may have been availed from the SHGs, continued membership for poor households ultimately requires investment in income-generating initiatives as SHG loans are not interest-free and require repayment.

The interaction of survey round with the loan purpose variable in this analysis revealed that households could *initially* be willing to view SHGs as a potential source of funds to meet health expenses, potentially encouraging them for joining SHGs and also encouraging them to stay l as members for longer compared to those who have not borrowed. However, the extent of this initial enthusiasm declines in Round II while the positive association continues. Availing loan for any purpose is positively associated with SHG membership and duration in both round I and round II, but in a lesser extent in Round II compared to Round I. The exact mechanism explaining this phenomenon is not well understood. However, as mentioned previously, the literature suggests that over-indebtedness is a recognized issue in microfinance, with any underlying borrowing reason, especially for health care expenses, being a stimulus for repeat borrowing thereby increasing household debt [9, 10, 38].

The effect of loan purpose among different gradients of the poor on SHG membership and duration was also evaluated in this study by interacting last loan purpose with wealth quintiles. The results showed that compared to the marginally poor households (quintile 1), poorer households in all quintiles (2, 3, 4 and 5), were less likely to *become SHG members* when they borrowed for health-related expenses, more likely when they borrowed for enterprise reasons. Among those poorer households (quintile 4), the odds of becoming a SHG member is 1.69 times higher (OR = 1.69, 95%CI:1.00–2.85, p<0.05), but their duration as a member on average 0.86 times lower (IRR: 0.86, 95%CI:0.75–0.99, p<0.05), when borrowing for enterprise reasons. Moreover, the likelihood of *continuing membership for longer* is higher in all household's wealth quintiles (2, 3, 4 and 5) when borrowing for health-related expenses and enterprise or income-generating activities, except in the fourth quintile households.

These preferences and associations indicate that in the rural Uttar Pradesh context, poorer households may be turning to SHGs to borrow money for health reasons which help them tide over health-related expenditures. Overall studies from India found that vulnerable households at the threshold of poverty get pulled deeper into poverty when exposed to maternal and newborn health issues due to lack of affordable financial sources to pay for treatment that is mainly Out of Pocket [39–42]. Moreover a study from India has shown that a woman's social affiliation has a strong influence on the extent of catastrophic maternal health care expenditure [41]. Therefore, households that receive financial reprieve through SHGs may be in a better economic situation to initially build household savings and reserves that permit them to participate in income-generating or enterprise activities as continuing members of SHGs.

## Effect of education on SHG membership and duration

Beyond economic indicators, this study also revealed that increasing levels of education of eligible women and their husbands had contrasting effects on SHG membership and duration. For women, being educated is seen to be associated with a higher likelihood of membership but not with duration, suggesting that more educated women may have a *greater* individual agency to try an initiative but choose not to continue if they do not derive benefit. Contrary to the effects of women's education, the results show women whose husbands are more educated were initially *less likely to be members*, but once they joined, they were significantly more likely to remain members. A likely explanation is that educated husbands are not initially supportive

of their wives in joining SHGs due to limited awareness or understanding of how membership may translate into household improvement. We see that levels of education have a differing effect on the duration of membership, suggesting differences in how more educated women do (or do not) utilise microfinance. Although this reasoning does not adequately explain why more educated husbands would prefer that their wives stay members as found in our study. One line of reasoning worth pursuing relates to more educated men leverage their wives to borrow funds for their businesses, or for purposes where the wife has little say on loan usage [13, 14, 38, 43, 44]. However, since the woman herself is the primary borrower in the SHG and the responsibility of loan repayment rests with the woman, an educated, empowered woman may be less willing to borrow funds in her name when her husband controls the use of borrowed funds as also observed in previous studies [13, 14, 38, 43].

Higher education for women in our study does not, however, play a protective role in the duration of membership, contrary to another study [15] from India, which found that women with some education were 30 per cent more likely to remain members and that SHGs, where all members were educated, were more likely to survive longer. In our study, women's secondary or higher education status was negatively associated with duration of the membership. This may be due to more educated women with higher education (a proxy for relatively better-off households) preferring non-microfinance-based sources for credit. Our result in this regard is supported from another study in India that found better-off households (representing higher educated members) preferred borrowing from informal credit sources over microfinance and were more comfortable in directly navigating the formalities associated with banks [13, 15].

## Impact of social class (caste) on SHG membership and duration

Previous studies from different states of India found that women belonging to scheduled caste were more likely to be members in SHG only programs [13, 15]. Our results support this evidence and show that membership for women belonging to scheduled caste (SC) is higher even in the Integrated SHG and health literacy program evaluated in this research. We, however, found that women belonging to scheduled caste and tribes are *less likely to continue longer as* SHG members (shorter average membership duration) as compared to general caste members.

While previous studies have found a similar association [13, 15], our study extends this association of reduced membership duration in rural Uttar Pradesh and in the context of an integrated microfinance and health literacy program. We also tested for the independent effect of caste alone on duration (results not reported here) using an appropriate count data model and found a significant negative effect of caste on the *duration of membership in SHG*. Studies elsewhere have also shown in some instances that upper caste members are more likely to *join* SHGs and remain members of the groups they joined [15]. Our study found the same effect for women in rural Uttar Pradesh with SC/ST and OBC household's membership duration almost 13 per cent *less* on average as compared to other general caste women. Among SC/ST castes, our study results are similar to previous evidence that found households belonging to Scheduled Tribes (ST) more likely to leave SHGs as compared to SC [15]. This is pertinent in Uttar Pradesh which predominantly has a larger SC than ST population, suggesting that community programs in UP may need strategies to support both SC and ST households for continuing as members [24]. It is also important to highlight that in the content of this study, IMFHL program implementation focussed on areas that had higher health and poverty burden. The average proportion of SC/ST population in the intervention and control blocks was quite similar (27 per cent and 29 per cent) [25]. While SC/ST is a critical parameter indicating low development, the measure is also used by the program and Indian Government alike to

identify marginalised households. In rural communities, caste-based hierarchies dictate social norms and are a contributing source of entrenched poverty for lower caste (SC/ST) households as they encounter greater social restrictions/barriers and marginalisation. The IMFHL project by design seeks to challenge these caste-based hierarchies by empowering women belonging to lower castes through microfinance participation. The program focus explains the higher SHG membership among SC/ST households, as seen in this analysis.

## Religion and SHG membership

We also evaluated another key social factor, religion, that governs many social norms in rural India. This study found that women from Hindu households were 1.19 times more likely to become SHG members as compared to Muslim households (HH). While Hindu HH compared to Muslims HH were less likely to *continue as* SHG members (lower membership duration), the association was not statistically significant. Religion as a determinant of membership and duration is important considering that community groups such as SHGs present opportunity to reach HH belonging to different minority religions (Islam, Sikhism, Christianity) in India. The largest minority religion in India, Islam comprise 13 per cent of the national population and 19 per cent of UP's population and in some instances has encountered higher social exclusion and deprivations [45–47]. Thus, India's current federal poverty alleviation program, the NRLM that uses a similar SHG model as evaluated in this paper presents an opportunity to focus on social policy and extend benefits of memberships in SHG program to all vulnerable communities.

## Maternal health indicators and SHG membership and duration

The SHG program implemented in rural UP and analysed in this paper also had an added maternal health literacy component. The current study found that an increasing number of previous pregnancies (parity) was associated with a higher likelihood of membership. While SHGs usually comprise of adult women in later part of their reproductive age group who may have completed their families, the inclusion of parity permits analysis of membership behaviours for younger members separate from the eligible woman's age. This finding contrasts with another study from a different part of India, which found that parity was negatively associated with membership, especially if the mother was already a carer of an under-five-year-old [15]. However, there is a fundamental program design difference between the previous study and the current study. While the previous study looked at a microfinance only *intervention*, our study, that is the IMFHL program investigates the effect of a microfinance intervention with an added maternal health literacy component. As observed in the IMFHL program, the higher membership association with parity may be due to the additive effect of the added health literacy component of the program.

The health literacy sessions added to regular SHG meetings offer women with maternal health knowledge and may be acting as an incentive for women in favour of SHG membership. SHGs by design provide a platform for discussion and knowledge sharing, and mothers may be more likely to join groups with the added value of a health literacy component. The peer effect in SHG groups other than access to credit, for example, was recently attributed by a study in India that showed increased school enrolment of members children as compared to non-members [48]. This is pertinent in the rural UP context where almost 70 per cent of women lack access to traditional sources of information through the media (television, radio, newspaper) coupled with gaps in the quality of health information provided through the public health system [49]. The membership appeal in SHGs for women and mothers can thus be understood. Moreover, a previous study from India showed an inverse relation of increasing

parity and health care utilisation, potentially re-exposing women with a higher number of pregnancies to maternal health risks [50]. Therefore, membership in SHGs and the associated peer effect may positively influence desired maternal health behaviours, especially among women with higher parity. Therefore, adding a health intervention to a microfinance program may incentivise membership for women who are also targets for maternal health promotion interventions.

While additional research is still required, our findings show the effects of layering health on microfinance and provide evidence to policymakers/advocates keen on adding health intervention to India's federal poverty alleviation plan, the N.R.LM which currently reaches 620 million households and present an opportunity to accelerate towards the 2030 Sustainable Development Goals for health and poverty reduction [51].

## Health service availability and microfinance

In the Indian health system, public health facilities are distributed according to population levels with the availability of primary health facilities indicative of rural residence or remoteness. [52]. Villages with any available health facility are considered better connected and therefore viewed as being less rural. In the program villages evaluated in our study, we found that women in villages with both government and private health facility had a longer duration of the membership. This may suggest that villages with health facilities may be less rural or better connected to roads and therefore more able to provide greater opportunity to engage in rural micro-enterprise for SHG members. This study also revealed that a higher number of community health workers in a village, and more contacts with them during pregnancy *increases membership likelihood*, as does having an institutional delivery. However, only the association for the number of community health worker with SHG membership is statistically significant. Other studies [11, 12] from the same project, previously reported that program implementation increased the social network and coordination between community health workers and the SHG platform leading to an increased density of social network and transmission of health information. This suggests a pathway of linking SHG membership and the number of community health workers in a village.

Our study differed from past studies in the methodological rigour employed to assess determinants of membership and duration simultaneously. Indeed, the hurdle negative binomial model employed in this study is better suited for a SHG based model where membership and duration are both considered in the same model but analysed separately. Moreover, no large-scale study in the context of rural India primarily rural Uttar Pradesh previously provided empirical evidence on the association between the poorest households and SHG membership/ duration, importantly in relation to the availability of BPL cards amongst various gradients of the poor. Additionally, this study fills a gap in the microfinance literature by triangulating the influence of poverty on membership and duration by using the availability of BPL cards, wealth quintile and different levels of the poor within BPL cardholders.

## Limitations

A couple of key limitations of the study are highlighted. Although the same villages were visited in both survey rounds, different households were sampled in both rounds. The lack of temporal data prevents examination of determinants of continued membership following the same woman/household in both survey rounds. Additionally, it is possible that a few households may have been captured in both survey rounds; however, these would be a limited sample. Another limitation is the possibility of recall bias, especially for women who were asked to remember details of the last loan taken. The value of loan amount which differentiates non-

borrowing and borrowing households that is seven USD (Rs 500) may miss including households that routinely borrow smaller amounts.

Another limitation is in examining area-level effects across villages while there may potentially be clustering within villages. As per the survey design, administrative Blocks were selected within districts, then GPs within blocks, then households within GPs. The sampling also allowed selecting control blocks matched roughly to intervention blocks, and then villages within blocks that are expected to be broadly similar across crucial developmental parameters. Consequently, the standard cluster sampling effects did not apply; there seemed to be more variations across villages than within villages. However, in India, rural health services and facilities are implemented at the level of the village, and they are likely to vary across villages. Therefore, it makes more sense to study the effect of area-level health system variables such as health facility availability, number of community health workers across villages on SHG membership and duration.

Further, the way the outcome variable of SHG membership was designed, sampled women comprising SHG members and Non-members, are selected from the same villages. However, in some cases, the women sampled in a village were either all members or all non-members, leading to outcome variables not exhibiting enough variations within a village. Therefore, it makes more sense to assess the determinants of outcome variables across villages instead of within villages as the clustering within villages may be less likely to have an influence here. Lastly, the study's cross-sectional design prevents concluding causality from observed relationships. The investigation of poor and rural women in the sampled districts of UP may prevent generalisation to greater UP or to other states of India that are culturally different. Further research, including studies with temporal data for households in rural India, are needed to assess determinants of membership duration in programs that integrate health and microfinance.

## Policy and program implications

Based on the study findings, policies may aim for suitability of the SHG model to meet the needs of the poorest households in these villages. It is possible that in its current form SHGs are only able to adequately meet the financial requirements of the poorest at the start of membership and are better suited to continue to cater to the poor below the poverty threshold. Efforts are required to communicate the benefit of SHG membership not only to the marginally poor but to all gradients of the poor, especially the poorest, religious minority and marginalised groups (Scheduled Castes/Tribes). More importantly, there is a need to develop better methods of capturing various gradients of the poor to improve beneficiaries targeting, selection, and retention into SHG programs.

Second, program planners and implementers may explore different communications strategies to inform potential members of the potential reasons for which loans can be availed from SHGs, especially for health reasons. One way that could improve borrowing loans for health reasons is to offer those loans at lower interest rates as compared to loans for enterprise and non-enterprise reasons. In the rural Indian context, the associations between limited access to funds, adverse health and deepening poverty are well established. Therefore, making it possible for households to take loans for health and illness may indirectly contribute to overall SHG functioning.

Finally, specifically focused qualitative research may help in exploring reasons on why rural poor women are joining SHGs but, not staying longer as members. This may inform strategies that support enrolled women to continue their membership with a view to improving overall, SHG member enrolment and membership duration.

## Conclusion

The SHG model analysed in this paper seeks to target the poorest and most marginalised households in one of India's most populated and developmentally disadvantaged states, Uttar Pradesh. Based on the findings, this study recommends that microfinance programs need to examine their inclusion and retention strategies in favour of poorest household using multidimensional indicators that are better able to capture poverty in its myriad forms. The addition of a maternal health literacy component along with access to funds for health reasons complements the underlying microfinance platform as members gain dual value from membership. Both components synergistically operating may also help to interrupt the mutually reinforcing cycle of poverty and poor maternal health. Evidence from a study in the state of Bihar, India also established the cost-effectiveness in promoting maternal and neonatal health when microfinance and health interventions are implemented at scale [53]. However, further research is required to capture costs effectiveness in promoting integrated microfinance and health interventions in Uttar Pradesh that is culturally distinct. This research also generated evidence on membership profiles of SHGs and provided insights for broader microfinance research elsewhere that may further evaluate the microfinance outcome.

## Supporting information

**S1 Appendix. Analysis framework: Determinants for SHG membership and duration for women in rural Uttar Pradesh, India.**
(TIF)

**S2 Appendix. Different count data models for SHG duration with goodness of fit measures.**
(DOCX)

**S3 Appendix. Summary statistics of selected variables by Non-Self-Help Group [Non-SHG] household and Self-Help Group [SHG] households.**
(DOCX)

**S4 Appendix. Results from the multivariate logit regression for determinants of SHG membership and hurdle negative binomial regression for determinants of SHG duration of membership.**
(DOCX)

## Acknowledgments

We wish to thank Professor Rachel Davey, Health Research Institute (HRI), the University of Canberra, for her guidance and reviewing the first draft of this manuscript. The authors would also like to thank the University of Canberra, the Public Health Foundation of India, and Population Council for actively supporting inter-organisational research collaboration.

## Author Contributions

**Conceptualization:** Danish Ahmad, Itismita Mohanty.

**Data curation:** Danish Ahmad, Itismita Mohanty.

**Formal analysis:** Danish Ahmad, Itismita Mohanty, Theo Niyonsenga.

**Funding acquisition:** Dileep Mavalankar.

**Investigation:** Danish Ahmad, Itismita Mohanty, Theo Niyonsenga.

**Methodology:** Danish Ahmad, Itismita Mohanty, Theo Niyonsenga.

**Project administration:** Danish Ahmad.

**Resources:** Itismita Mohanty, Laili Irani, Dileep Mavalankar.

**Supervision:** Itismita Mohanty, Dileep Mavalankar, Theo Niyonsenga.

**Validation:** Danish Ahmad, Itismita Mohanty, Laili Irani, Dileep Mavalankar, Theo Niyonsenga.

**Visualization:** Danish Ahmad, Itismita Mohanty.

**Writing – original draft:** Danish Ahmad.

**Writing – review & editing:** Danish Ahmad, Itismita Mohanty, Laili Irani, Theo Niyonsenga.

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
