## [Decision Letter · Decision Letter 0]

12 Mar 2020

PONE-D-20-01535

Participation in Micro-finance based Self Help Groups in India: Who becomes a Member and for How Long

PLOS ONE

Dear Danish Ahmad,

Thank you for submitting your manuscript to PLOS ONE. After careful consideration, we feel that it has merit but does not fully meet PLOS ONE’s publication criteria as it currently stands. Therefore, we invite you to submit a revised version of the manuscript that addresses the points raised during the review process.

 AS ACADEMIC EDITOR, I have made comments in the response below.

We would appreciate receiving your revised manuscript by Apr 26 2020 11:59PM. To enhance the reproducibility of your results, we recommend that if applicable you deposit your laboratory protocols in protocols.io, where a protocol can be assigned its own identifier (DOI) such that it can be cited independently in the future. For instructions see: http://journals.plos.org/plosone/s/submission-guidelines#loc-laboratory-protocols

We look forward to receiving your revised manuscript.

Kind regards,

Eileen Stillwaggon

Academic Editor

PLOS ONE

Additional Editor Comments (if provided):

We were pleased to review this well-written and important article and look forward to the revision. The reviewers both indicated Minor Revision, which I think is an indication of their confidence that the authors can satisfactorily address all of their concerns. I share that confidence, but I have upgraded the decision to Major Revision to emphasize the importance of their comments. Both reviewers have done a thorough job and I am pleased that the journal allowed them extra time to complete the reviews. Reviewer 1: I believe that the authors must already have the additional information requested. The reviewer's comments are an invitation to strengthen the article substantially by enriching the background about the participants, and other comments. In addition to the nice summary at the beginning, I count 12 distinct points that Reviewer 1 would like to see addressed, as well as the last paragraph about the tables. I urge the authors to read the comments carefully -- for my own reading, I found it useful to mark them up and separate them into separate tasks -- and address as many as possible. As always, the authors do not have to agree with every point, but they do need to respond to every point.

Reviewer 2 has also done a very thorough read and the comments appear in an annotated copy that hopefully will be attached to this response. If you do not receive Reviewer 2's annotated manuscript, please contact the journal to get it. Please respond to all comments, although as said, the authors do not have to agree to every change, if they deem the change incorrect.

Journal Requirements:

2. Please include additional information regarding the survey or questionnaire used in the study and ensure that you have provided sufficient details that others could replicate the analyses. For instance, if you developed a questionnaire as part of this study and it is not under a copyright more restrictive than CC-BY, please include a copy, in both the original language and English, as Supporting Information. In addition, please provide any details of pre-testing of the questionnaire.

Reviewers' comments:

Reviewer's Responses to Questions

**Comments to the Author**

1. Is the manuscript technically sound, and do the data support the conclusions?

Reviewer #1: Partly

Reviewer #2: Yes

2. Has the statistical analysis been performed appropriately and rigorously? 

Reviewer #1: Yes

Reviewer #2: Yes

3. Have the authors made all data underlying the findings in their manuscript fully available?

Reviewer #1: No

Reviewer #2: Yes

4. Is the manuscript presented in an intelligible fashion and written in standard English?

Reviewer #1: Yes

Reviewer #2: Yes

5. Review Comments to the Author

Reviewer #1: This paper utilizes logistic and hurdle negative binomial regression analyses to model Self-Help Group (SHG) membership and duration in the Integrated Microfinance and Health Literacy (IMFHL) program in Uttar Pradesh in India in 2013 and 2015. Using a range of economic, sociodemographic, and area-level characteristics, the authors find that asset-based wealth quintiles, which measure the gradients of poor households, may be a stronger indicator of SHG membership and duration than income. Results suggest that women from the lowest wealth quintile are less likely to become members, unless they borrow for health reasons. However, they are likely to continue as members for a longer period if they borrow funds for income-generating activities.

This paper seeks to address research gaps in identifying and analyzing the determinants of microfinance membership and duration involving women-only SHG in rural India. A significant contribution of the paper is that it utilizes asset-based wealth quintiles to identify various degrees of poverty and better investigate the effectiveness of microfinance program in reaching the poorest households. Given that women-only SHGs have been popularly adopted in South Asia as a poverty alleviation tool, this is a policy-relevant topic. Furthermore, the paper’s investigation of who benefits from such groups and what drives participation in these groups help address the role of microfinance programs in alleviating poverty.

Overall the paper is well-developed and well-written. The authors do a commendable job in providing detailed background information on SHGs in India, which help contextualize them with regards to the social and microfinance programs designed to help the poorest households in the country. However, the paper includes limited information about the women participants in the SHGs. To provide a better understanding of how the program works, it would be important to know who these women were and what criteria was used to determine their eligibility in the groups. Given that ‘member selection… is a crucial consideration for the succession of such programs’ (pg. 5), additional information about whether or not these women were from the same community and already knew each other before participation, or whether they were familiar with the SHG program could provide important insights into their engagement with the program.

The authors conduct a thorough analysis of the important determinants of SHG membership and duration in rural India. Although the survey data is collected from eligible women in both IMFHL program and non-program households (pg. 8), information on what factors were used to determine eligibility for the survey is lacking. It is not clear whether the factors were the same as those used for SHG eligibility, and if so why. Moreover, it is concerning that the paper lacks information on how the control group was created. There is little discussion about the characteristics of women in the control group and how these characteristics were important for the study. Given that women who chose to participate in SHG programs can be inherently different from those who did not, control group needs to be properly defined to avoid any selection bias issues. Although the summary statistics (Table 2) highlight the similarities and differences between the treatment and control groups for some relevant variables, the paper should provide more information about what factors were used to create these groups.

I am also concerned about the potential for multicollinearity issues in the model due to inclusion of variables that might be correlated with each other. For example, the variable working status of eligible women for cash, kind, or both, a proxy for household poverty, could be correlated to women’s education levels. If women with higher education were more likely to be working relative to their less educated counterparts, this may bias their impact on SHG membership and duration. Additionally, if households with educated partners were less likely to be poor relative to those with lower levels of education, it would differently impact their participation in the program. It would be interesting to see how interactions between the education levels and other relevant variables might impact the results of the study. Some robustness checks would be useful to ensure the results are not biased due to multicollinearity.

Furthermore, the paper would benefit from a detailed discussion of some of its results. An interesting finding is that, accounting only for loan purposes, when women borrowed funds for health reasons they were less likely to become SHG members but more likely to continue their membership for a longer period. However, when asset-based wealth quintiles were included in the analysis, the results suggested that women in the lower wealth quintiles were most likely to become SHG members when they borrowed for health reasons, but would likely continue the membership for longer if they borrowed for income-generating activities only. The paper does not include enough discussion and clarity on this seemingly contradicting results on the impact of loan purpose on membership and duration. Similarly, the results show that women belonging to Scheduled Caste were more likely to be members of the program compared to those from the majority Hindu general caste. Given that caste system is highly prevalent in the Indian society, it would be interesting if the authors could discuss this relationship in the context of SHGs and its implications on credit accessibility to socially marginalized groups.

The findings of the study provide important insights into factors that determine participation in SHG programs and the duration of membership. However, the results may need to be presented in a clear way for readers to better understand the implications of this study. Details regarding the suitability of the various count data models and Table 1 (pg. 13) could be included in the Appendix. Summary statistics presented in Table 2 (pg. 14-16) is highly informative. However, a shorter version of the table would suffice to highlight the key similarities and differences between the treatment and control groups. Moreover, Table 3 in its current form is too long and difficult to follow. I recommend that the authors include a shorter version of the table such that it highlights the main results. More detailed versions of Tables 2 and 3 can be included in the Appendix.

Reviewer #2: Well written paper on an important topic. Suggest few minor changes, most important being in the conclusion. Have attached the manuscript with my comments for your consideration. The tables also needs some review.

6. PLOS authors have the option to publish the peer review history of their article (what does this mean?). If published, this will include your full peer review and any attached files.

Reviewer #1: No

Reviewer #2: No

---

## [Author Response · Author response to Decision Letter 0]

18 May 2020

We thank the editor and the reviewers for the comments as we think it adds value to the paper and improves the overall presentation. We have also provided a track change version of the manuscript, however, for ease of reference, the page and line numbers indicated in the response to reviewers document refer to the cleaned version of the manuscript.

---

## [Decision Letter · Decision Letter 1]

20 Jul 2020

PONE-D-20-01535R1

Participation in Micro-finance based Self Help Groups in India: Who becomes a Member and for How Long

PLOS ONE

Dear Dr. Ahmad,

Thank you for submitting your manuscript to PLOS ONE. After careful consideration, we feel that it has merit but does not fully meet PLOS ONE’s publication criteria as it currently stands. Therefore, we invite you to submit a revised version of the manuscript that addresses the points raised during the review process.

We look forward to receiving your revised manuscript.

Kind regards,

Srinivas Goli, Ph.D.

Academic Editor

PLOS ONE

Additional Editor Comments (if provided):

Both reviewers agree that the revised version of the paper is an improved version over its previous version and recommended for publication. However, reviewer 3 suggested a few corrections which need to be implemented before recommending it for publication.

Reviewers' comments:

Reviewer's Responses to Questions

**Comments to the Author**

1. If the authors have adequately addressed your comments raised in a previous round of review and you feel that this manuscript is now acceptable for publication, you may indicate that here to bypass the “Comments to the Author” section, enter your conflict of interest statement in the “Confidential to Editor” section, and submit your "Accept" recommendation.

Reviewer #2: All comments have been addressed

Reviewer #3: All comments have been addressed

2. Is the manuscript technically sound, and do the data support the conclusions?

Reviewer #2: Yes

Reviewer #3: Yes

3. Has the statistical analysis been performed appropriately and rigorously? 

Reviewer #2: Yes

Reviewer #3: Yes

4. Have the authors made all data underlying the findings in their manuscript fully available?

Reviewer #2: Yes

Reviewer #3: Yes

5. Is the manuscript presented in an intelligible fashion and written in standard English?

Reviewer #2: Yes

Reviewer #3: Yes

6. Review Comments to the Author

Reviewer #2: The authors have addressed the comments from the reviewers and revised version is much improved and is acceptable for publication.

Reviewer #3: REVIEW PONE-D-20-01535R1

Title: Participation in Micro-finance based Self Help Groups in India: Who becomes a Member and for How Long

Abstract:

The abstract presents determinants of membership and staying a member in an integrated microfinance and health literacy program from one of India’s poorest and most populated states, i.e. Uttar Pradesh through varied explanatory variables in terms of socio-economic, demographic and area related characteristics.

Methods: The paper utilises secondary survey data from the Uttar Pradesh Community Mobilization project comprising of 15,300 women from SHGs and Non-SHG households in rural India and performed multivariate logistic and hurdle negative binomial regression analyses to model SHG membership and duration.

Result:

The paper covers detailed analysis of the specialised microfinance model involving women Self-Help Groups in rural India with an additional layer of health intervention. The SHG model is popular and prevalent in South Asia and India adopted the SHG program as part of its national poverty alleviation program and is home to a substantial, large poor population. The authors have very pertinently provided detailed background information on SHGs in India, which is helpful in contextualizing with regards to the social and microfinance programs designed to help the poorest households in the country.

Very pertinently Authors has taken care of reviewers comments step by step and finally the outcome in the form of paper (revised version) is in good shape. But there are certain issues which I would like to point out as following:

186 development regions of Uttar Pradesh (55). Under the IHLMF program, is wrongly put, it should be (IMFHL) Integrated Microfinance and Health Literacy

190 plus health (intervention blocks) in the IHLMF program.is wrongly put and should be (IMFHL) Integrated Microfinance and Health Literacy

215 (out of a total of 80 districts in U.P state). U.P. is administratively subdivided into 80 districts is wrong as Uttar Pradesh is comprised of 75 districts. (Uttar Pradesh is divided into 75 Districts and 18 Divisions.Mar 2, 2020 Uttar Pradesh District Map, List of Districts in Uttar Pradesh)

www.mapsofindia.com

If these corrections are inserted I think the paper has touched upon a very important issue of poverty eradication through SHG policy in the State of Uttar Pradesh.

I strongly recommends the publication of the paper.

7. PLOS authors have the option to publish the peer review history of their article (what does this mean?). If published, this will include your full peer review and any attached files.

Reviewer #2: No

Reviewer #3: No

---

## [Author Response · Author response to Decision Letter 1]

21 Jul 2020

We thank the editor and the reviwers for their valuable feedback and for the recommendation to publish after addressing these changes.

---

## [Editor Report · Decision Letter 2]

22 Jul 2020

PONE-D-20-01535R2

Participation in Micro-finance based Self Help Groups in India: Who becomes a Member and for How Long

PLOS ONE

Dear Dr. Ahmad,

Thank you for submitting your manuscript to PLOS ONE. After careful consideration, we feel that it has merit but does not fully meet PLOS ONE’s publication criteria as it currently stands. Therefore, we invite you to submit a revised version of the manuscript that addresses the points raised during the review process.

ACADEMIC EDITOR: Look at my comments at the bottom. 

We look forward to receiving your revised manuscript.

Kind regards,

Srinivas Goli, Ph.D.

Academic Editor

PLOS ONE

Additional Editor Comments (if provided):

No doubt your paper is making significant contributions and it will be recommended for publication on behalf of me for sure. However, can you consider the following minor comments from my end?

Can you check the reference citation carefully? It should be chronological in order. The first cited reference should be first. Why reference no. 55 is coming much ahead of its order.

Also, please cite the most suitable references keeping in mind the context of the study. Do you refer to all the existing literature in the context of Uttar Pradesh relevant to your subject? I found some sloppiness in this case.

For instance, take the following sentence:

“Overall studies from India found that vulnerable households at the threshold of poverty get pulled deeper into poverty when exposed to maternal and newborn health issues due to lack of affordable financial sources to pay for treatment that is mainly Out of Pocket (41, 42).”.

Does any of the following papers assess newborn health issues and OOPE? One is focused on institutional deliveries and the other is on prenatal and natal care. Both suffer from significant data and methodological limitations. Moreover, none of them are in the Uttar Pradesh context. Considering health systems are vary considerably across the states, OOPE and Catastrophic OOPE also vary across the states.

[41] Govil D, Purohit N, Gupta S, Mohanty S. Out-of-pocket expenditure on prenatal and

natal care post-Janani Suraksha Yojana: a case from Rajasthan, India. Journal of Health,

Population and Nutrition. 2016; 35(1).

[42] Mohanty S, Srivastava A. Out-of-pocket expenditure on institutional delivery in India. Health Policy and Planning. 2012; 28(3):247-262 1010 https://academic.oup.com/heapol/article/28/3/247/551401

Thus, try to cite more relevant literature for the sentence that your writing. Why do you ignore other more comprehensive studies which considered greater details of background characteristics and also assessed postnatal care too? I have given some references as below which overcome the several data and methodological limitations compared to the studies you have cited. You can refer to them if you would like to.

Goli, S., Rammohan, A., and Pradhan, J., 2016. High spending on maternity care in India: What are the factors explaining it?. PloS one, 11(6), p.e0156437.

Goli, S., and Rammohan, A., 2018. Out-of-pocket expenditure on maternity care for hospital births in Uttar Pradesh, India. Health economics review, 8(1), p.5.

These are just an example, but you need to check other references too.

I am giving one more reference which is a recent comprehensive study that documents on “Poverty, Employment, Health and Education, and Access to various Social Safety nets” across Six Social Religious Groups. You can refer to it if you would like to.

Kumar S, Prashan F, Trivedi K, Goli S. BACKWARD AND DALIT MUSLIMS: Education, Employment and Poverty.

---

## [Author Response · Author response to Decision Letter 2]

26 Jul 2020

We thank the editor for sharing additional resources and for their valuable feedback. The references shared add value to the claims made in the paper and we are thankful to the editor for highlighting these articles which are now included.

---

## [Editor Report · Decision Letter 3]

29 Jul 2020

Participation in Micro-finance based Self Help Groups in India: Who becomes a Member and for How Long

PONE-D-20-01535R3

Dear Dr. Ahmad,

We’re pleased to inform you that your manuscript has been judged scientifically suitable for publication and will be formally accepted for publication once it meets all outstanding technical requirements.

Kind regards,

Srinivas Goli, Ph.D.

Academic Editor

PLOS ONE